# IFNγ-Producing B Cells Play a Regulating Role in Infection-Mediated Inhibition of Allergy

**DOI:** 10.3390/biology12091259

**Published:** 2023-09-20

**Authors:** Sai Qiao, Ying Peng, Chunyan Zhang, Rony Thomas, Shuhe Wang, Xi Yang

**Affiliations:** 1Department of Immunology, Rady Max College of Medicine, University of Manitoba, Winnipeg, MB R3E 0T5, Canada; saiqiao@zju.edu.cn (S.Q.); xiaoying1919@gmail.com (Y.P.); wfchuanqi@126.com (C.Z.); thomasr8@myumanitoba.ca (R.T.); shuhe.wang@umanitoba.ca (S.W.); 2Department of Medical Microbiology and Infectious Diseases, Rady Max College of Medicine, University of Manitoba, Winnipeg, MB R3E 0T5, Canada; 3Department of Clinical Laboratory, Sir Run Run Shaw Hospital, School of Medicine, Zhejiang University, Hangzhou 310016, China

**Keywords:** B cells, chlamydial infection, asthma, IFNγ

## Abstract

**Simple Summary:**

The hygiene hypothesis suggests that some infections may inhibit the development of allergic diseases, but the mechanism remains unclear. This research demonstrated the important role of cytokine-producing B cells in the infection-mediated modulation of allergic responses. Using chlamydia as a model pathogen, we showed that the adoptive transfer of B cells isolated from chlamydia-infected mice, unlike those from naïve mice, could effectively inhibit allergic airway eosinophilia and mucus overproduction, as well as Th2 cytokine responses. Intracellular cytokine analysis showed that B cells from chlamydia-infected mice produced higher levels of IFNγ than those from naïve mice. The inhibitory effect of the adoptively transferred B cells on allergic reactions was virtually abolished by the simultaneous blockade of IFNγ using a monoclonal antibody, which suggested that B cells modulated by chlamydial lung infection could have an inhibitory effect on airway allergic responses via the production of IFNγ. The results provide new insights into the targets related to the prevention and treatment of allergic diseases.

**Abstract:**

The hygiene hypothesis suggests that some infections may inhibit the development of allergic diseases, but the mechanism remains unclear. Our previous study has shown that Chlamydia *muridarum* (Cm) lung infection can inhibit local eosinophilic inflammation induced by ovalbumin (OVA) through the modulation of dendritic cell (DC) and T cell responses in mice. In this study, we explored the role of B cells in the chlamydial-infection-mediated modulation of allergic responses. The results showed that adoptive transfer of B cells isolated from Cm-infected mice (Cm-B cells), unlike those from naïve mice (naïve B cells), could effectively inhibit allergic airway eosinophilia and mucus overproduction, as well as Th2 cytokine responses. In addition, total IgE/IgG1 and OVA-specific IgE/IgG1 antibodies in the serum were also decreased by the adoptive transfer of Cm-B cells. Intracellular cytokine analysis showed that B cells from Cm-infected mice produced higher levels of IFNγ than those from naïve mice. More interestingly, the inhibiting effect of adoptively transferred Cm-B cells on allergic reactions was virtually abolished by the simultaneous blockade of IFNγ using a monoclonal antibody. The results suggest that B cells modulated by chlamydial lung infection could play a regulatory role in OVA-induced acute allergic responses in the lung via the production of IFNγ. The results provide new insights into the targets related to the prevention and treatment of allergic diseases.

## 1. Introduction

The notion that atopy and allergy can be inhibited by infection was initially introduced by Strachan in 1989 [1]. His classical study found that first-born children showed a higher frequency of allergic rhinitis and atopic dermatitis than their siblings, likely due to the lower possibility of the former being exposed to common infections. Since then, many epidemiological and experimental studies have provided evidence supporting the hygiene hypothesis that is extended to explain the increases in other inflammatory and autoimmune diseases [2,3,4]. Many microorganisms, including bacteria, viruses, and parasites, are found to be inhibitory for allergic responses in experimental models and clinical settings [5,6,7,8]. Our previous research found that chlamydial and mycobacterial infections can inhibit antigen-specific allergic responses induced by OVA and ragweed sensitization and challenge in mice [5,9]. Although significant progress has been made in this area, the underlying mechanisms by which infections modulate allergic diseases remain elusive.

Asthma is an airway inflammatory disorder characterized by hyperresponsiveness, local inflammation, structure remodeling, mucus hypersecretion, angiogenesis, smooth muscle hypertrophy, and sub-basement membrane fibrosis. OVA is a commonly used model allergen to stimulate an allergic reaction in animal experiments. The features of OVA-induced airway acute allergic responses include eosinophils infiltration, mucus hypersecretion, and increased Th2 cytokines, as well as OVA-specific IgE in the serum and lung. Our previous studies have demonstrated that chlamydial and mycobacterial infections can switch the allergen-specific CD4+ T cells from Th2- to Th1-like cells in the OVA and ragweed sensitization and challenge models [5,9]. In particular, we found that Cm-infected mice showed significantly increased IFNγ production by OVA-specific T cells after OVA sensitization and challenge [5]. We also found that these infections can enhance regulatory T cell (Treg) responses [10]. The alterations in T cell responses are associated with the inhibition of allergic reactions. Furthermore, we found that the changes in T cell responses are largely reflected by the alteration of dendritic cell (DC) function [11]. In addition, we found that chlamydial-infection-induced NK cells play an important role in the infection-mediated inhibition of allergic responses induced by OVA [12]. The results suggest that multiple types of immune cells are involved in the process of infection-mediated inhibition of allergies.

Traditionally, the function of B cells is considered to contribute to adaptive immunity by secreting antibodies and working as antigen-presenting cells (APCs). IgE antibody production by B cells in allergy is a critical component of the pathological process. However, recent studies suggested that B cells can also produce cytokines, such as IL-10, IL-17, and IFNγ, contributing to the modulation of immune responses [13]. In particular, IL-10-producing B cells were found to function like regulatory cells, earning them the name regulatory B cells (Breg) [14]. In addition, IL-10-producing B cells were found to contribute to host defense against infections. For example, in primary Coxiella *burnetii* infection, B cells defend against infection by secreting high levels of IL-10 [15]. According to the differential secretion of IFNγ or IL-4, B cells may be subgrouped into two populations which may subsequently regulate naïve CD4+ T cells differentiating to Th1 or Th2 cells [16]. Although the studies on cytokine-producing B cells are still limited, the available data thus far suggest that, in addition to their role in the production of antibodies, B cells may play other roles in the process between the interaction of infection and allergy.

To more specifically address the mechanism underlying infection-mediated regulation of allergy, we used an adoptive transfer strategy to test the influence of B cells activated by Cm infection on allergic reactions in a mouse model of asthma-like lung inflammation. We found that B cells isolated from chlamydia-infected mice could inhibit OVA-induced asthma-like inflammatory responses. The inhibitory function of Cm-B cells is associated with their IFNγ production. Further study using simultaneous IFNγ-blocking antibodies confirmed the contribution of IFNγ produced by Cm-B cells in the inhibition process. The finding suggests that IFNγ-producing B cells play a significant role in the infection-mediated inhibition of allergic reactions and imply that this type of B cell may be involved in a broader spectrum of immune regulation.

## 2. Materials and Methods

### 2.1. Mice

Male C57BL/6 mice were used for the study at 6–8 weeks of age; approximately 100 mice were sacrificed for the research. The mice were bred and maintained at a pathogen-free animal care facility at the University of Manitoba, Canada. All experiments were performed following the guidelines issued by the Canadian Council of Animal Care. The Animal Ethical Committee of the University of Manitoba approved the animal protocol (No. 15-008).

### 2.2. Organism

Chlamydia *muridarum* (Cm) organisms (Nigg strain) were cultured, purified, and quantified, as reported in our previous study (5). In brief, Cm was cultured in HeLa-229 cells using RPMI 1640 medium supplemented with 10% fetal bovine serum (FBS), 1% L-glutamine, and 25 mg/mL gentamycin. The elementary bodies (EBs) were purified and collected by discontinuous density gradient centrifugation. EB infectivity was measured by infecting Hela-229 and immunostaining chlamydial inclusions. The purified EBs were suspended in sucrose–phosphate–glutamic acid buffer and stored at −80 °C. The same batch of purified EBs was used throughout this study.

### 2.3. Chlamydia Infection and B Cell Isolation

Naïve mice were intranasally (i.n.) inoculated with 1 × 10^3^ inclusion-forming units (IFU) of Cm EBs. Ten days post-infection, the spleens of naïve mice and Cm-infected mice were aseptically removed, and B cells were isolated using MACS (Miltenyi Biotech, Bergisch Gladbach, Germany) B220 beads according to the manufacturer’s instructions. Briefly, spleen single-cell suspension was prepared in PBS with 0.5% BSA. MACS B220 beads were used for the positive selection of B220+ cells using the columns. Purified B cells were used for adoptive transfer through the tail vein or i.n. routes. As specified in the experiments, the transfer of B cells was performed with or without an anti-IFNγ antibody (BD Pharmingen, San Diego, CA, USA) or isotype control IgG antibody (BD Pharmingen). The purity of the isolated B cells was 95% to 99%, based on flow cytometry analysis of B220+ cells (Appendix A). Intracellular cytokine analysis showed that B220+ cells were virtually the sole cell population that produced IFNγ (Appendix A). Considering that B220+ cells may contain cells other than B cells, we stained the splenic B220^+^ cell with CD19, CD11c, and NK1.1 antibodies and found that over 97% of the cells were CD19+ B cells (Appendix A).

### 2.4. Allergen Sensitization/Challenge and Cell Adoptive Transfer

Mice were initially sensitized intraperitoneally (i.p.) with 2 μg of OVA and 2 mg Al(OH)_3_ adjuvant in 100 μL HBSS. On day 14 post-sensitization, mice were challenged intranasally with 50 μg OVA in 40 μL HBSS (Appendix A). For adoptive transfer experiments, 2 h before sensitization and challenge, B cells (2 × 10^6^ cells/mouse) isolated from naïve or Cm-infected mice were injected through the tail vein. The other steps were the same as in the routine OVA challenge/sensitized model (Appendix A). For antibody-blocking experiments, mice were sensitized i.p. with 2 μg OVA and 2 mg Al(OH)_3_ in 40 μL HBSS. At 14 days post-sensitization, mice were intranasally administered naïve B cells or Cm-B cells (1 × 10^6^ cells/mouse) with or without anti-IFNγ antibody (10 μg) or isotype control IgG antibody (10 μg) in 40 μL HBSS. Two hours later, mice were challenged intranasally with OVA (50 μg/mouse) in 40 μL HBSS (Appendix A). All mice were euthanized 5 days later and analyzed for allergic inflammation and immune responses.

### 2.5. Bronchoalveolar Lavage Fluids and Leukocyte Differentials

The main trachea of a mouse was cannulated, followed by a wash of the lung with 1 mL of PBS two times to collect bronchoalveolar lavage (BAL) fluids. Cells in the BAL fluids were counted and prepared for smears using a Cytospin machine (Thermo, Waltham, MA, USA). The slides were air-dried and then stained with Hema-3 Stain kit (Fisher Scientific, Hampton, NH, USA), which contained a cell fixative and eosin Y stain. The number of eosinophils, monocytes/macrophages (mon/Mφ), and lymphocytes per 200 cells were counted and differentiated based on cellular morphology and staining characteristics. Cells in the BAL were also analyzed by flow cytometry (Canto II, BD) for eosinophils with fluorescence-conjugated mAbs.

### 2.6. Preparation of Lung and Spleen Single-Cell Suspensions

Lung and spleen single-cell suspensions were prepared for flow cytometry analysis. The lungs were collected from mice and digested with 2 mg/mL collagenase XI (Sigma-Aldrich, St. Louis, MO, USA) in RPMI 1640 for 1 h at 37 °C; 20% EDTA was then added at 5 min before the digestion ended. Then, red blood cells were lysed using ACK lysis buffer (150 mM NH_4_Cl, 10 mM KHCO_3_, 0.1 mM EDTA) and the cell suspensions were filtered. All the cells were washed and resuspended with FACS buffer (PBS without Ca^2+^ and Mg^2+^ containing 2% FBS and 0.09% NaN_3_). The spleens were ground using a 70 μm cell strainer for spleen cells, then lysed with ACK buffer, followed by centrifugation and resuspension with PBS buffer.

### 2.7. Analysis of Lung Pathology

The lung tissues of mice were collected and fixed in 10% buffered formalin. Tissue sections were stained with H&E (hematoxylin and eosin) for histology. Periodic-acid Schiff (PAS) Staining Kit (Sigma Aldrich) was used for bronchial mucus and goblet cell testing, and the staining slides were examined and quantified for a histologic mucus index (HMI) [9].

### 2.8. Cell Culture and Cytokine Analysis

Mice were euthanized on day 5 post-OVA challenge and examined for cytokine production by both lung and spleen cells. The lung and spleen single-cell suspensions were prepared aseptically. They were cultured at a relative concentration of 7.5 × 10^6^ cells/mL (spleen) and 5.0 × 10^6^ cells/mL (lung) with OVA (0.4 mg/mL) in the complete RPMI 1640 medium containing 10% FBS, 25μg/mL gentamycin, 2 mM L-glutamine, and 0.05 mM 2-mercaptoethanol. Culture supernatants were harvested at 72 h for the measurement of cytokines by ELISA.

### 2.9. Immunoglobulin Determination in Blood Serum

ELISA was used for analyzing total and OVA-specific IgE and IgG1 Abs in the serum. Sera were determined for OVA-specific IgG1 Abs using biotinylated goat anti-mouse IgG1 Abs (Southern Biotechnology Associates, Birmingham, AL, USA). For determination of total and OVA-specific IgE, sera were first incubated with a 50% slurry of GammaBind G Sepharose (GE Healthcare Life Science, Piscataway, NJ, USA) to remove most of the serum IgG and then were measured for OVA-specific IgE Abs using an ELISA kit (BioLegend, San Diego, CA, USA). Total IgG1 and IgE were measured by ELISA with purified and biotinylated antibodies (Southern Biotechnology Associates).

### 2.10. Flow Cytometry Analysis

For cell surface molecule staining, cells were blocked with anti-CD16/CD32 Abs (eBioscience, San Diego, CA, USA) for 20 min followed by surface molecule staining using anti-CD45-PerCp (BD Pharmingen), anti-Siglec-F-PE (eBioscience), anti-CD11c-FITC (eBioscience), and Fixable Viability Dye-eFlour 506 mAbs (eBioscience). After incubating on ice in the dark for 40 min, the cells were washed and resuspended with a fix buffer (eBioscience) for flow cytometry analysis. For intracellular cytokine staining, lung and spleen single-cell suspensions were cultured at concentrations of 5 × 10^6^ cells/mL and 7.5 × 10^6^ cells/mL, respectively, in the complete RPMI 1640 medium with Brefeldin A Solution (eBioscience), which includes PMA, ionomycin, and BFA, for 5 hrs. After incubation, cells were collected and blocked with anti-CD16/32 Abs for 20 min and stained with anti-B220-FITC (BD Pharmingen), anti-CD3e-PE-Cy7 (eBioscience), and Fixable Viability Dye-eFlour 506 mAbs (eBioscience) for 40 min. Cells were then fixed and washed in permeabilization buffer and stained with anti-IFNγ-APC, IL-10-APC, or IL-4-APC (eBioscience), or with isotype control Abs, for 40 min. Cells were washed twice with permeabilization buffer and analyzed by flow cytometry, which was based on the gating strategy shown in Appendix A.

### 2.11. Statistical Analysis

Data were statistically analyzed using Student *t*-tests, one-way ANOVA, two-way ANOVA, and Turkey’s multiple comparisons test (GraphPad Prism 5 software); *p* < 0.05 was considered significant. All the experiments were repeated three or more times. In each repeating experiment, every group included 3–4 mice.

## 3. Results

### 3.1. Adoptive Transfer of B Cells from Cm-Infected Mice Attenuates OVA-Induced Airway Eosinophilic Inflammation

We first checked the effect of Cm-B cells on OVA-induced asthma-like reactions by the adoptive transfer of Cm-B cells to syngeneic mice that were then sensitized and challenged with OVA. The influence of Cm-B cells on local allergic reactions was assessed by histological and cellular analyses. H&E and PAS staining of lung tissue sections revealed that the transfer of Cm-B cells dramatically diminished the eosinophilic inflammation (Figure 1A) and the mucus overproduction (Figure 1D). In contrast, the adoptive transfer of B cells from naïve mice (naïve B cells) showed little effect on the allergic reactions, similar to the PBS control group. Consistently, the BAL fluid cell analysis showed that the recipients of Cm-B cells only had minimal infiltrating inflammatory cells, unlike the recipients of naïve B cells and PBS (Figure 1B,C). Cytologically, the minimal infiltrating cells in Cm-B recipients were mainly macrophages rather than allergic eosinophils that dominated the groups of naïve B cell recipients and PBS controls (Figure 1C). Quantitation of mucus production by histological mucus index (HMI) measurement also showed dramatic mucus reduction in the Cm-B recipients (Figure 1D). Together, the data suggest that B cells from chlamydia-infected mice play an important role in the infection-mediated inhibition of airway allergic inflammation induced by OVA.

### 3.2. Cm-B Cells Alter the Pattern of Cytokine Response and Immunoglobin Production Induced by Allergen

IL-4 and IL-5 are cytokines that play a major role in allergic eosinophilic inflammation. To elucidate the basis for reducing the inflammation caused by Cm-B cell transfer, we tested the levels of these cytokines in the Cm-B cell recipients compared with naïve B cell recipients and the PBS-treated group. We found that in both the lung (Figure 2A) and the spleen (Figure 2B), the levels of IL-4 and IL-5 in the recipients of Cm-B cells were dramatically lower compared with the recipients of naïve B cells and the PBS control group. We also found a dramatic decrease in total IgE (Figure 2C) and OVA-specific IgE (Figure 2D) levels in the sera of Cm-B cell recipients. In addition, the levels of total IgG1 (Figure 2E) and OVA-specific IgG1 (Figure 2F) were also dramatically reduced in the Cm-B recipients. However, the level of IgG2a did not show significant changes by the cell transfer. Further analysis of allergen-driven IL-4 production by CD4+ T cells showed that Cm-B cell recipients exhibited lower IL-4 production compared to the recipients of naïve B cells and PBS group (Figure 3). All these results suggest that Cm-B cells are inhibitory for Th2-related cytokines and antibody isotypes.

### 3.3. Cm-B Cells but Not Naïve B Cells Produce Higher Levels of IFNγ

Having demonstrated the inhibitory role of Cm-B cells in OVA-induced airway eosinophilic inflammation and Th2 cytokine production, we further explored the capacity of Cm-B cells in producing cytokines related to immune regulation, including IL-4, IL-10, IL-17, and IFNγ, by intracellular cytokine staining. Of the cytokines measured, IFNγ was the only one that was found to be produced at a significantly higher level by the Cm-B cells isolated from the spleen and the lung than the naïve B cells (Figure 4A,B). The B cells producing other cytokines (IL-4, IL-10, and IL-17) were comparable between Cm-infected and naïve mice (Figure 4A,B).

### 3.4. The Inhibitory Effect of Adoptively Transferred Cm-B Cells on OVA-Induced Airway Allergic Reactions Is Abolished by Simultaneous Blockade of IFNγ Function

To directly test whether IFNγ produced by Cm-B cells was the key factor relevant to the inhibitory effect of these cells on OVA-induced eosinophilic inflammation, we adoptively transferred Cm-B cells with simultaneous anti-IFNγ mAb administration. We found that the inhibitory effect of Cm-B cells on airway allergic inflammation was virtually abolished by the co-delivery of anti-IFNγ mAb, while the co-administration of isotype control IgG antibody had no such reversal effects (Figure 5A). The reversal effect of anti-IFNγ mAb was also confirmed by the analysis of BAL fluids for inflammatory cells, showing increased absolute numbers (Figure 5B) and the percentage of eosinophils (Figure 5C). A similar reversal effect was seen for mucus production (Figure 5D). Notably, the administration of anti-IFNγ mAb to the mice without Cm-B cell transfer had no significant effect on the allergic inflammation (Figure 5A–C) and mucus production (Figure 5D) induced by OVA sensitization and challenge. Flow cytometry analysis confirmed the reversal effect of anti-IFNγ mAb on Cm-B-cell-mediated eosinophilia reduction (Figure 6). The results suggest that the IFNγ produced by Cm-B cells plays a critical role in the inhibitory function of these cells on airway allergic inflammation.

### 3.5. The Inhibitory Effect of Cm-B Cells on Allergen-Driven Th2-Related Cytokines and Antibodies Was Abolished by co-Administration of anti-IFNγ mAb

To further test the role of IFNγ production by Cm-B cells in allergen-driven Th2 cytokine production, we examined the IL-4 and IL-5 production by the mice that received ex vivo splenic and pulmonary Cm-B cell transfer with co-administration of anti-IFNγ mAb or isotype control IgG antibody. We found that the significant inhibitory effect of Cm-B cells on IL-4 and IL-5 production was largely reversed by the co-delivery of anti-IFNγ mAb while the co-delivery of isotype control antibody had no significant effect (Figure 7A,B), which was consistent with the effect on allergic eosinophilic inflammation. The sole administration of anti-IFNγ mAb without Cm-B cell transfer had no effect on the levels of these cytokines in the OVA-sensitized and OVA-challenged mice (Figure 7A,B). A similar reversal effect on total (Figure 7C) and OVA-specific (Figure 7D) IgE and IgG1 (Figure 7E,F) antibodies by the anti-IFNγ mAb co-administration was also found. The reversal effect of the anti-IFNγ mAb co-administration on IL-4 production by allergen-driven CD4+ T cells was also confirmed by intracellular cytokine staining (Figure 8). The results suggest that the inhibitory effect of Cm-B cells on airway allergic inflammation, Th2 cytokine production, and IgE and IgG1 antibody responses is largely dependent on their IFNγ production.

## 4. Discussion

This study extended our research on the mechanism of Cm-infection-mediated inhibition of asthma-like reactions by examining the role of B cells in this process. Using an adoptive transfer approach, we showed that B cells from infected mice have the capacity to dramatically inhibit allergic reactions. Specifically, adoptive transfer of B cells isolated from Cm-infected mice significantly inhibited airway eosinophilia and mucus overproduction, reduced Th2 cytokines production, and allergic asthma-related total and allergen-specific antibody responses. More interestingly, we found that the B cells from Cm-infected mice produce IFNγ, which is likely the key mechanism for the modulatory effect of these cells on allergic reactions. The results suggest that IFNγ-producing B cells may play a critical role in the modulating effect of infections on allergic responses.

Aside from its classical role in antibody production, the immunoregulatory function of B cells has been gradually appreciated in recent years. Among the multiple potential mechanisms involving the effect of B cells in immune modulation, the role of regulatory B cells, which produce a large amount of IL-10, is most widely recognized. Indeed, IL-10-producing Breg cells (B10) have been shown to play an important role in the inhibition of autoimmune diseases and inflammations. For example, with the loss of B10 cells, older Tim-1-mutant mice were found to develop spontaneous autoimmune responses associated with hyperactive T cell expansion [17]. Similarly, during a viral brain infection, B10 cells could directly modulate T lymphocyte and microglial cell responses as well as promote CD4^+^ Foxp3^+^ T regulatory cell proliferation to control neuroinflammation [18]. The inhibitory role of Breg cells in allergic diseases has also been reported. In a cockroach-allergen-induced airway inflammation model, the deficiency of regulatory B cells in mice led to increased allergic airway inflammation with a higher level of Th2 cytokines [19]. In the house-dust-mite-induced murine asthma model, a CD9^+^ Breg cell subset was identified as the major inhibitor of the asthmatic responses [20]. Moreover, Amu et al. found that regulatory B cells from helminth-infected mice could reverse established airway inflammation via inducing regulatory T cells [21]. Therefore, our initial exploration was also more focused on identifying IL-10-producing Breg cells induced by Cm infection. However, after extensive exploration, we found that the role of classical Breg appeared not significant in this model because the intracellular cytokine staining of Cm-B cells and the culture of spleen cells from infected mice did not show a meaningful increase in IL-10 production. Instead, we found a close association between the inhibitory effect of Cm-B cells and their IFNγ production. Indeed, we found that among the cytokines related to immune modulation in allergic reactions, IFNγ is the only cytokine showing a significant increase in production by the B cells isolated from Cm-infected mice (Figure 4). More importantly, we found that the simultaneous neutralization of IFNγ at the time of Cm-B cell adoptive transfer virtually abolished the modulating effect of these cells on allergic reactions. The data, together with the previous reports on the modulating role of B cells, suggest that, depending on the type of infection or disease, B cells may play their regulatory role through different cytokine signaling pathways. In particular, in infections of intracellular bacteria which often bias a Th1-like response, such as chlamydia, B cells may more likely modulate immune responses through IFNγ, rather than IL-10 production. Notably, this is not the first report that shows the role of IFNγ-producing B cells in immune-related diseases [22,23,24,25,26].

It is reported that IFNγ-producing B cells contributed to the pathogenesis of proteoglycan-induced arthritis (PGIA) [24,27]. In addition, treatment of collagen-induced arthritis (CIA) with agonistic anti-CD40 antibody reduced IFNγ production by B cells, leading to improvement of the disease [26]. Notably, all these reports showed a pathogenic role of IFNγ-producing B cells. However, in our model, IFNγ-producing B cells showed a protective role in allergy and asthma. However, considering that the IFNγ-producing B cells only occupy of small portion of the transferred B cells, we think that the IFNγ production of B cells is more likely a mechanism for these B cells to modulate other cells that are related to the development and promotion of allergic reactions. In other words, IFNγ likely needs to be delivered to the right cells in the right niche, possibly through cell–cell interactions. The details of the potential cell–cell interaction and the manner of cytokine delivery by B cells and consequent signaling pathways in this process is subject to future study. On the other hand, the protective role of IFNγ-producing B cells by promoting Th1 cells in infectious diseases has been well documented [28,29,30]. Therefore, the opposite role played by B cells observed in this and previous studies [24,26,27] is likely due to the nature of the corresponding diseases, i.e., Th2-related asthma and Th1-related autoimmune diseases.

It might be of concern that we used B220 as a major marker for identifying B cells in the study, because B220 also reportedly expresses on non-B cells, such as NK and dendritic cells (DCs). To ensure that the observation in this study is a reflection of B cell function, we analyzed the percentage of different types of cells, including B cells, NK cells, and DCs, in the B220+ population based on their CD19, CD11c, and NK1.1 markers. The results showed that more than 97% of the B220+ cells are CD19+ cells, i.e., B cells. Moreover, the minimal population of NK1.1+ cells and CD11c+ B220+ cells showed no significant changes in Cm-infected mice in comparison with naïve mice (Appendix A). In the adoptive transfer experiments, a clear inhibition of OVA-induced asthma-like responses was found even with the transfer of B220+ cells with nearly 99% purity. The results strongly suggest a significant role of B cells in inhibitory function. More importantly, our intracellular cytokine staining of the purified cells showed that the minimally contaminated B220− cells virtually had no IFNγ production (Appendix A). It could be of concern that the transferred B cells from infected mice might carry live Cm, which infects the recipient mice, thus modulating T cell responses because the infection is not fully clear at day 10. However, we tested and did not see live Cm in the isolated splenic B cells. This is likely because Cm mainly infects local (lung) epithelial cells and macrophages. Another potential concern is the possible contamination of cells other than B cells in the transferred B cell preparation, because some T cells, DCs, and NK cells reportedly also express B220. In particular, because IFNγ-producing Th1 cells have been shown to inhibit allergic reactions, the involvement of the contaminated cell in the process needs to be addressed. Therefore, we carefully analyzed the purity of the B cell preparation and the capacity of the contaminated non-B cells for IFNγ production. As shown in Appendix A, the purity of the B cell preparation was over 97% (Appendix A), and the CD3^+^ cell produced a negligible amount of IFNγ, which is unlikely to impact the effect of B cell transfer for the inhibition of allergic response (Appendix A). Moreover, very few non-B cells were found in the isolated B220+ cells (Appendix A). Taken together, we feel confident that the observed inhibitory effect of the adoptive transfer of B cells on allergic reactions in the study is mainly due to the modulatory role of the IFNγ-producing B cells.

## 5. Conclusions

In summary, the present study reveals an important role played by B cells in chlamydial-infection-mediated modulation of allergic asthma-like inflammations. Moreover, the data suggested that the B cells from Cm-infected mice may play an immunomodulatory role through their IFNγ production. The identification of an immunoregulatory IFNγ-producing B cell population in infected mice provides new insights into the interaction between infection and allergy and, more broadly, immune regulation mechanisms.

## Figures and Tables

**Figure 1 biology-12-01259-f001:**
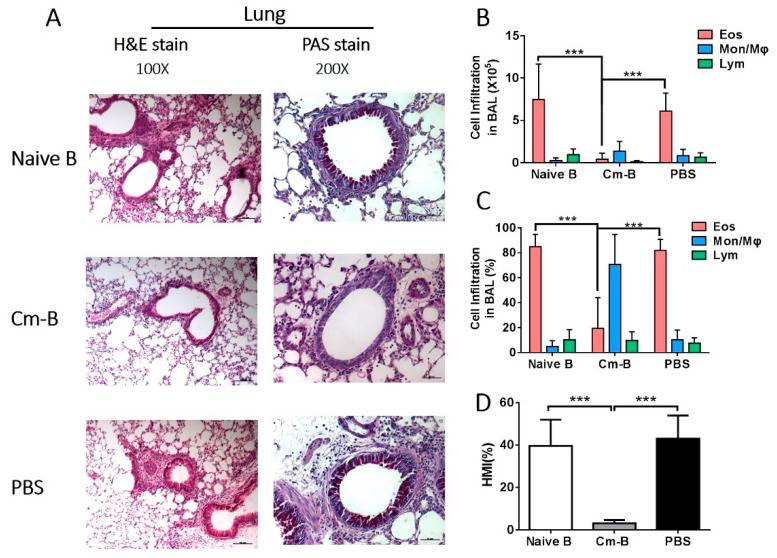
The effect of adoptive transfer of B cells on pulmonary inflammation, bronchial mucus production, and airway eosinophilia induced by OVA sensitization/challenge. B cells from naïve (Naïve B cells) or Cm-infected (Cm-B cells) mice were adoptively transferred to C57BL/6 mice through the tail vein at the time of OVA sensitization and challenge as described in Methods. Mice were sacrificed on day 5 post-OVA challenge. (**A**) Lung sections (5 μm) were stained with H&E (left panel, magnification of 100) or PAS (right panel, magnification of 200). The absolute number (**B**) and the proportion (**C**) of each cellular component composing the total BAL cells. (**D**) Histological mucus index (HMI) was calculated based on the percentage of the mucus-positive area over the total area of the airway epithelium. The experiment was repeated more than three times. Each group included three or more mice in each repeating experiment. The data shown are Mean ± SD. *** *p* < 0.001.

**Figure 2 biology-12-01259-f002:**
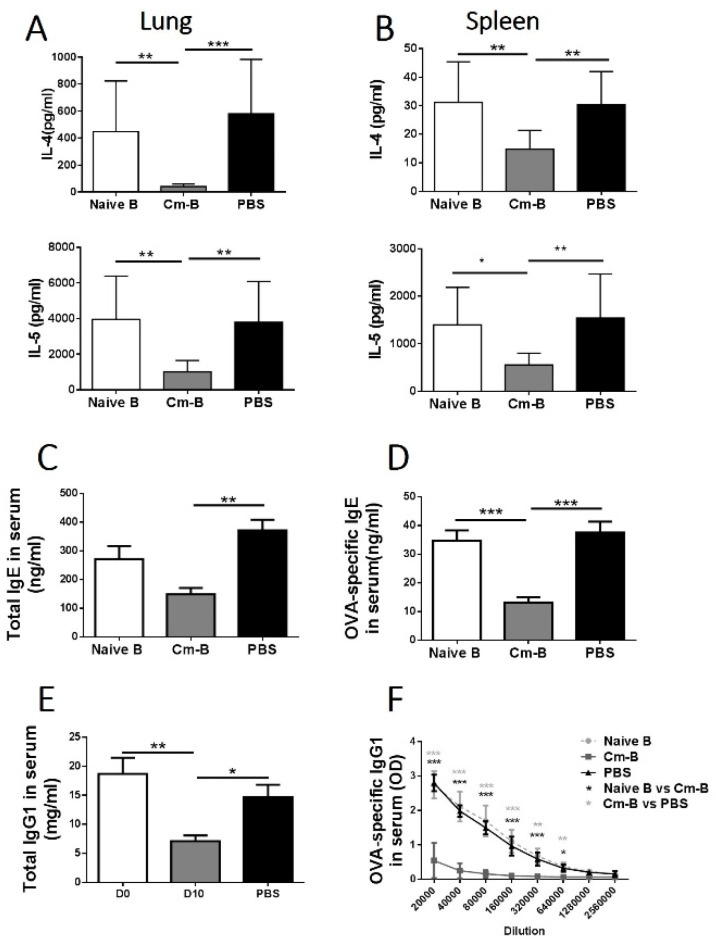
Dramatic reduction in OVA-driven Th2 cytokines and IgE/IgG1 antibodies in mice which received Cm-B cells. Mice were treated as described in Figure 1. Lung (**A**) and spleen (**B**) mononuclear cells were cultured in the presence of OVA. The culture supernatants were harvested after 72h incubation. Cytokines in the culture supernatant were detected by ELISA. Total IgE (**C**) and OVA-specific IgE (**D**), as well as total IgG1 (**E**) and OVA-specific IgG1 (**F**) in sera, are shown. The experiment was repeated more than three times. Each group included three or more mice in each repeating experiment. The data shown are Mean ± SD. * *p* < 0.05; ** *p* < 0.01; *** *p* < 0.001.

**Figure 3 biology-12-01259-f003:**
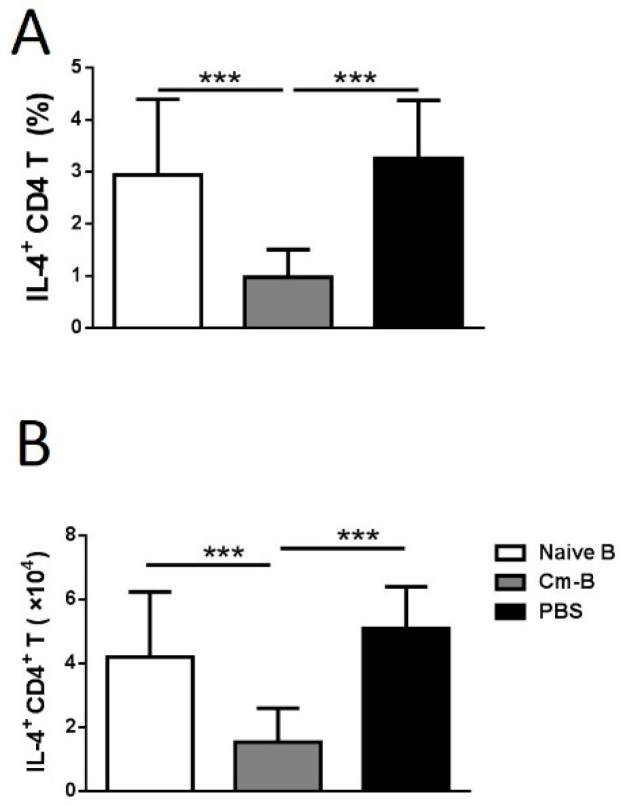
Adoptive transfer of Cm-B cells inhibited allergen-driven IL-4-producing CD4+ T cells. Mice were treated as described in Figure 1 and sacrificed for lung cell analysis on day 5 post-challenge with OVA intranasally. Intracellular IL-4 was detected using flow cytometry. The percentage (**A**) and absolute number (**B**) of IL-4-producing CD4+ T cells in the lungs were calculated. The experiment was repeated more than three times. Each group included three or more mice in each repeating experiment. The data shown are Mean ± SD. *** *p* < 0.05.

**Figure 4 biology-12-01259-f004:**
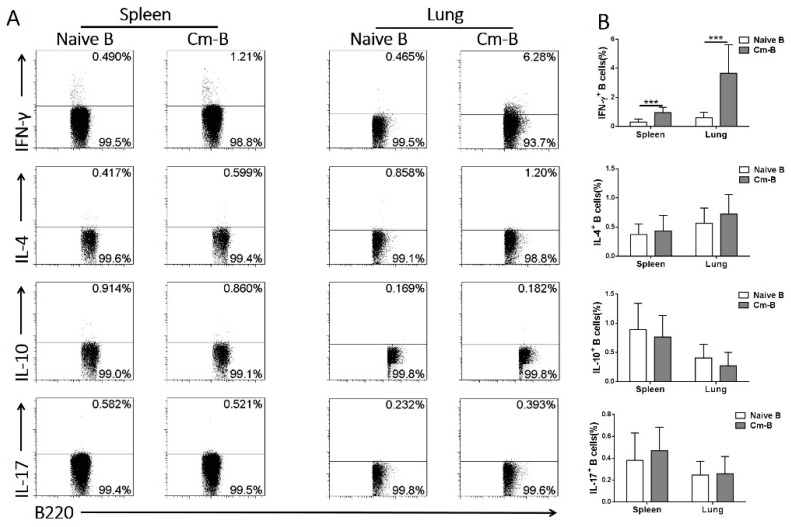
Cm-B cells produced higher levels of IFNγ than naïve B cells in both the spleen and lung. Mice were infected with Cm intranasally and were sacrificed on day 10 post-infection. (**A**) Intracellular cytokines of CD3^−^B220^+^ cells (B cells) from naïve and infected mice in the spleen and lung. (**B**) Summary data are shown as the mean ± SD of each group. The experiment was repeated more than three times. Each group included three or more mice in each repeating experiment. The data shown are Mean ± SD. *** *p* < 0.001.

**Figure 5 biology-12-01259-f005:**
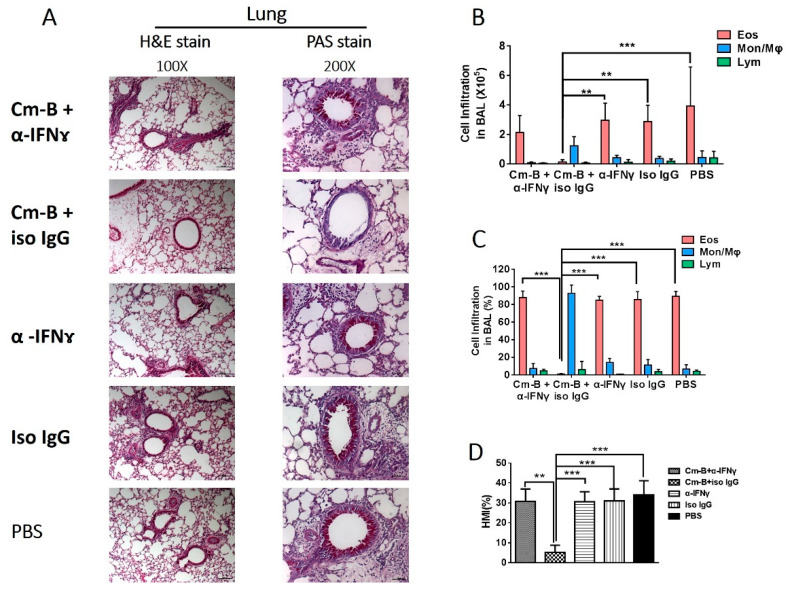
Blockade of IFNγ at the time of adoptive transfer abolished the inhibitory effect of Cm-B cells on OVA-induced asthma-like responses. Mice were sensitized i.p. with OVA and were challenged intranasally with OVA as described in Methods. Two hours before the intranasal OVA challenge, mice in the different groups were adoptively transferred with Cm-B cells plus anti-IFNγ (Cm+α-IFNγ), Cm-B cells plus antibody isotype control (Cm+iso IgG), anti-IFNγ alone (α-IFNγ) or isotype control alone (Iso IgG), or PBS alone, respectively. The magnification of H&E and PAS stain were 100 and 200, respectively. On day 5 post-challenge, mice were sacrificed for further analysis. (**A**) Lung sections were stained with H&E (left panel) or PAS (right panel). The absolute number (**B**) and percentage (**C**) of each cellular component composing the total BAL cells are shown. (**D**) HMI is calculated based on the percentage of the mucus-positive area over the total area of the airway epithelium. The experiment was repeated more than three times. Each group included three or more mice in each repeating experiment. The data shown are Mean ± SD. ** *p* < 0.01; *** *p* < 0.001.

**Figure 6 biology-12-01259-f006:**
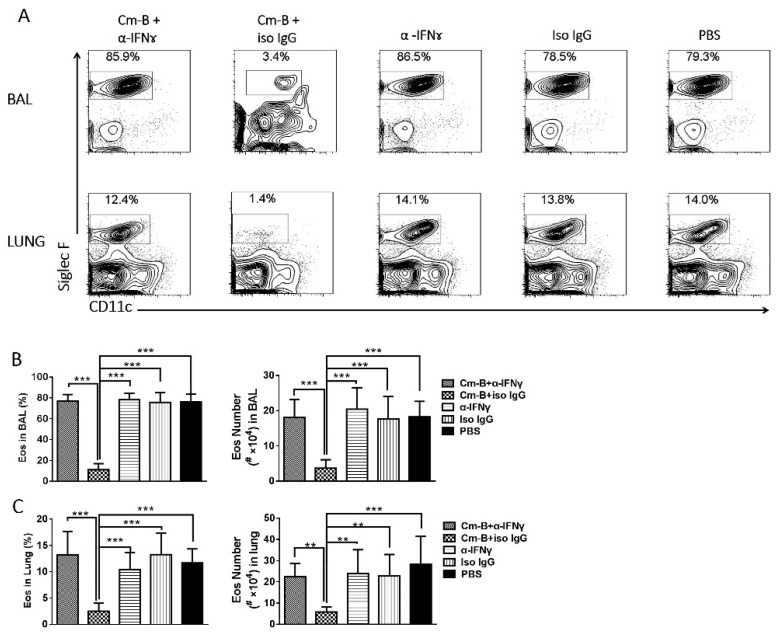
Flow cytometric analysis of eosinophilic responses in the lung tissues and BAL fluids following Cm-B cell transfer and anti-IFNγ antibody treatment. Mice were treated as described in Figure 5. (**A**) Single cells in BAL fluids and digested lung tissues were analyzed by flow cytometry. Cells were gated for live cells, and CD45+ cells and the CD11c-Siglec F+ cells were defined as eosinophils. The percentage (left) and the absolute number (right) of eosinophils in the BAL (**B**) and the lung (**C**) were shown as the mean ± SD of each group. The experiment was repeated more than three times. Each group included three or more mice in each repeating experiment. The data shown are Mean ± SD. ** *p* < 0.01; *** *p* < 0.001.

**Figure 7 biology-12-01259-f007:**
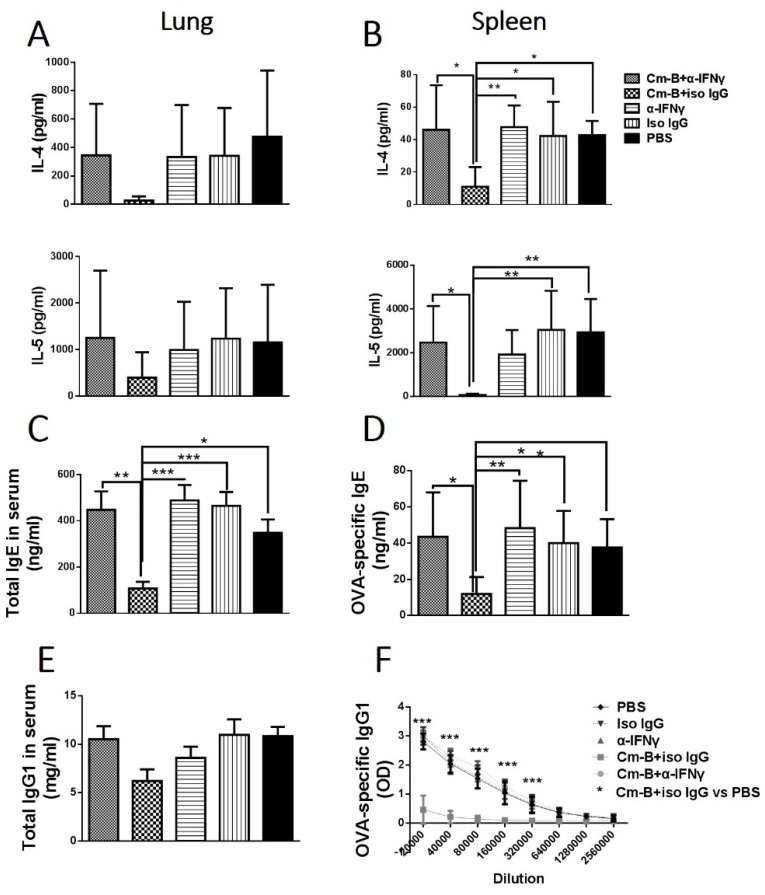
Blockade of IFNγ abolished the inhibitory effect of Cm-B cells on Th2 cytokines and OVA-specific IgE/IgG1 antibodies. Mice were treated as described in Figure 5. Lung (**A**) and spleen (**B**) mononuclear cells were cultured in the presence of OVA. The culture supernatants were harvested after 72 h incubation. Sera of each group were collected for measuring total IgE (**C**) and OVA-specific IgE (**D**), as well as total IgG1 (**E**) and OVA-specific IgG1 (**F**). Data are shown as mean ± SD of each group. The experiment was repeated more than three times. Each group included three or more mice in each repeating experiment. The data shown are Mean ± SD. * *p* < 0.05; ** *p* < 0.01; *** *p* < 0.001.

**Figure 8 biology-12-01259-f008:**
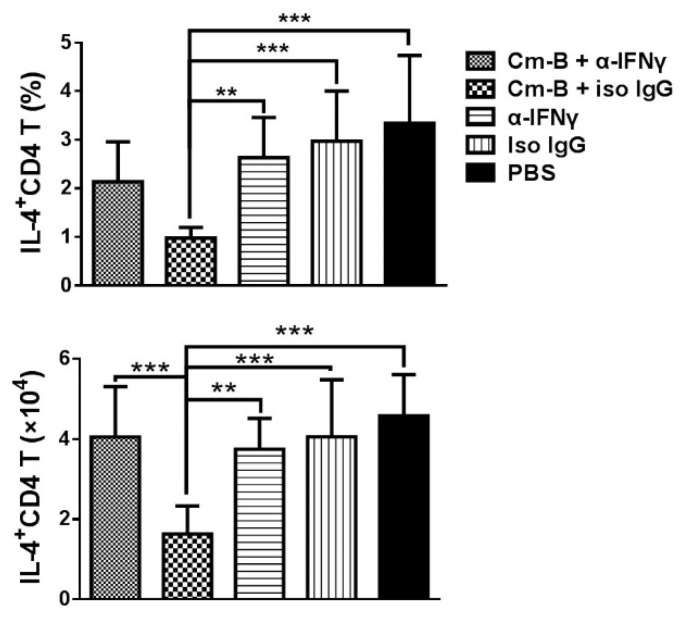
Neutralization of IFNγ at the time of cell adoptive transfer abolished the inhibitory effect of Cm-B cells on IL-4-producing CD4+ T cells. Mice were treated. Intracellular IL-4 was detected using flow cytometry. The percentage and absolute number of IL-4-producing CD4+ T cells in the lung were calculated. Data are shown as the mean ± SD of each group. The experiment was repeated more than three times. Each group included three or more mice in each repeating experiment. The data shown are Mean ± SD. ** *p* < 0.01; *** *p* < 0.001.

## Data Availability

Data was available by contacting authors by email.

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
