# Peer review of "IFNγ-Producing B Cells Play a Regulating Role in Infection-Mediated Inhibition of Allergy"

_biology, 2023, doi:10.3390/biology12091259_

Round 1

Reviewer 1 Report

The authors analyzed the role of B cells in the chlamydial infection-mediated modulation of allergic responses using Chlamydiae-infected mice and comparing them to B cells from naive mice and controls. Their results showed that adoptive transfer of B cells from Chlamydiae-infected mice inhibit allergic eosinophilia, mucus production and Th2 cytokine responses. They also demonstrated higher levels of IFN-gamma produced by B cells in Chlamydiae-infected mice, suggesting that B cells modulated by chlamydial lung infection could have a regulatory function in airway allergic responses via the production of IFN-gamma.

The investigation has been appropriately structured and conducted, with the support of statistical analysis.

However, there are some aspects that may need attention:

1. The data described in the text for the "Materials and Methods" section on "Chlamydiae infection and B cell isolation" refers to Supplementary Figure 1c (line 128, page 3) which does not match the informations presented as there is no Figure 1c.

2. In Figures 1 and 5, panel A shows low power microphotographs of lung tissue (H&E and PAS stains) and it is very difficult to appreciate the differences in eosinophilic infiltration and histological features described in the text. It might be more informative to use high power field photographs (in Figure 1) or even consider leaving out the Panel A with lung histology for Figure 5 since in this form it does not contribute to the illustration of the data described.

3. Minor typographical error: In the section "Materials and Methods", paragraph "Immunoglobins determination in blood serum" (line 180, page 5) should be replaced by "immunoglobulins".

Reviewer 2 Report

Qiao et al report that IFNg-production by B cells from chlamydia-infected mice mitigate ovalbumin-induced airway allergy in mice.  Thus, adoptive transfer of B cells from Cm-infected mice into mice followed by sensitization reduces inflammatory infiltrate, reduces IL-4/5 (Th2 responses), and lowers serum levels of OVA-specific IgE.  The authors examine IFNg+ B cells and find the frequency is increased in Cm-immunized mice.  In addition, neutralization of IFNg in recipient mice restores allergic responses, suggesting that B cell-derived IFNg contributes to the reduced response.

The experiments are clear, the data generally support the author’s conclusions, and the authors are careful not to overinterpret their results.  That said there are a few points to address:

1.    B cells used for adoptive transfer were isolated 10d following Cm challenge.  Is infection cleared by this timepoint?  The possibility that Cm (or parts thereof) may be transferred should be evaluated and/or discussed, especially considering immunization drives a Th1 response.

2.    Immunization with Cm leads to enhanced Th1 responses.  Though a relatively low frequency of B220+ cells immunostain for other cell types (i.e. DC and NK cells), it appears plausible that other types of cells, namely Th1 cells, could be adoptively transferred.  Consistent with this possibility, Figure 4a shows that the B220low cells stain for the highest amounts of IFNg.

3.    Related, magnetic bead separation of B cells resulted in ~95% purity; of the other 5%, it will be important to know how many T cells (e.g. Th1) are included.

4.    The authors state that 1-way ANOVA with Student’s t-test is used throughout.  However, there is more than one variable in most of the assays, necessitating a 2-way ANOVA.  Moreover, a multiple comparison test should be applied to compare groups within each experiment.

5.    The methods section lacks description of adoptive transfer.  In addition, supplementary figure 2 appears to indicate that multiple cell transfer approaches were used.  The approach needs to be clearly defined for individual figures.

minor edits are required as some terms read as plural when they are singular

Author Response

Please see the attachment, thank you!

Reviewer 3 Report

Qiao., et al. presented IFNγ producing B cells play a regulating role in infection-mediated inhibition of allergy. Using a Chlamydiae as a model pathogen authors showed that adoptive transfer of B cells isolated from Cm-B mice can inhibit allergic eosinophilia and mucus overproduction. These are quite interesting, and this manuscript deserves publication with minor revision.

1.     As a part of characterization of the system, evidence that Cm-B mice has produced more IFN gamma compared to naïve mice should be shown prior to presenting differences in the expression of other genes. (At transcription and translational level).

2.     The discussion is long, not cohesive, and is not presented in the context of the model.

3.     Line 128- The authors mentioned about supplementary figure 1C, which is not available.

4.     In Materials and Methods section – Cell culture and cytokine analysis, it is not clear what concentration of 2-mercaptoethanol was used (line-178).

5.     Figure 6 A labeling is not done properly (Example – Iso IgG.)

6.     Please check supplementary materials section as Table S1 is missing.  

7.     The whole manuscript needs to be edited to a consistent writing. It is hard to read and follow. Please, make appropriate changes.

The whole manuscript needs to be edited to a consistent writing. It is hard to read and follow.

Author Response

Please see the attachment, thank you!

Reviewer 4 Report

In this research article, the authors investigated the effect of IFNγ producing B cells derived from infection on autoimmunity in a mouse model. This is a rather elegant work, with sound experimental design. The conclusions are supported by the results presented in the study. This experimental design showed a correlation between infection and B-cell-mediated allergies in a relatively short timeframe. It would be interesting to explore whether exposure to pathogens at a young age could affect the development of autoimmunity later in life, to showcase the translatability of the findings.

Specific points:

Simple summary and abstract: In both sections, the significance of the study should be more apparent. Additionally, an introductory sentence on the simple summary should be added.

Lines 88-95: I would suggest avoiding an extensive description of the results of the study since this is the ending paragraph of the introduction.

Lines 98-102: The number of mice used should be added, as well as the approval number for the study given by the ethical committee overseeing this project.

Line 104: all bacterial names should be italicized.

Lines 155-165 and 190-204: these sections should be merged.

Lines 208-209, 239-240, 265-266, 272-273, 287-289, 318-319, 326-327, 354-355, 361-362: how many groups of 3-4 mice were included in the experimental design? How do the authors define how many times an experiment was repeated (number of animals, number of groups etc.)? Also, in figure legends, the authors state that: “One representative experiment of three independent experiments is shown.” This wording is vague. Additionally, it is also stated that “Data are shown as the mean ± SD of each group.” This suggests that the bars have resulted from the mean of all experimental repetitions which contradicts the previous statement. Appropriate adjustments should be made in the text.

Author Response

Please see the attachment, thank you!

Reviewer 5 Report

The manuscript is written and organized well. The paper shows the critical role of B- cells in the production of interferon-gamma and their role in infection-mediated modulation of allergic inflammations. 

The results are well organized and state the point of the manuscript. 

However, some figures can be improved, like in figure 3 legend- describe figure 3A  and 3B separately like in other figures. 

Figures 2F and 7D- you can use colors to represent different conditions, the figure is not very clear in black & white. 

Figure 6A- Labelling can be improved, you can't see the data clearly.

line 79 - spelling of defense is wrong.

I feel that the authors can add some more references. 

Overall, the manuscript is of significance.

Author Response

Please see the attachment, thank you!
